# ZeroNVS: Zero-Shot 360-Degree View Synthesis from a Single Real Image

## Abstract

We introduce a 3D-aware diffusion model, ZeroNVS, for single-image novel view synthesis for in-the-wild scenes. While existing methods are designed for single objects with masked backgrounds, we propose new techniques to address challenges introduced by in-the-wild multi-object scenes with complex backgrounds. Specifically, we train a generative prior on a mixture of data sources that capture object-centric, indoor, and outdoor scenes. To address issues from data mixture such as depth-scale ambiguity, we propose a novel camera conditioning parameterization and normalization scheme. Further, we observe that Score Distillation Sampling (SDS) tends to truncate the distribution of complex backgrounds during distillation of 360-degree scenes, and propose "SDS anchoring" to improve the diversity of synthesized novel views. Our model sets a new state-of-the-art result in LPIPS on the DTU dataset in the zero-shot setting, even outperforming methods specifically trained on DTU. We further adapt the challenging Mip-NeRF 360 dataset as a new benchmark for single-image novel view synthesis, and demonstrate strong performance in this setting.

## 1 Introduction

Models for single-image, 360-degree novel view synthesis (NVS) should produce *realistic* and *diverse* results: the synthesized images should look natural and 3D-consistent to humans, and they should also capture the many possible explanations of unobservable regions. This challenging problem has typically been studied in the context of single objects without backgrounds, where the requirements on both realism and diversity are simplified. Recent progresses rely on large datasets of high-quality object meshes like Objaverse-XL (Deitke et al., 2023) which have enabled conditional diffusion models to produce photorealistic images from a novel view, followed by Score Distillation Sampling (SDS; Poole et al., 2022) to improve their 3D consistency. Meanwhile, since image diversity mostly lies in the background, not the object, the ignorance of background significantly lowers the expectation of synthesizing diverse images–in fact, most object-centric methods no longer consider diversity as a metric (Liu et al., 2023b; Melas-Kyriazi et al., 2023; Qian et al., 2023).

Neither assumption holds for the more challenging problem of zero-shot, 360-degree novel view synthesis on real-world scenes. There is no single, large-scale dataset of scenes with ground-truth geometry, texture, and camera parameters, analogous to Objaverse-XL for objects. The background, which cannot be ignored anymore, also needs to be well modeled for synthesizing diverse results.

We address both issues with our new model, ZeroNVS. Inspired by previous object-centric methods (Liu et al., 2023b; Melas-Kyriazi et al., 2023; Qian et al., 2023), ZeroNVS also trains a 2D conditional diffusion model followed by 3D distillation. But unlike them, ZeroNVS works well on scenes due to two technical innovations: a new camera parametrization and normalization scheme for conditioning, which allows training the diffusion model on multiple diverse scene datasets, and a new "SDS anchoring" mechanism, addressing the limited background diversity from standard SDS.

To overcome the key challenge of limited training data, we propose training the diffusion model on a massive mixed dataset comprised of all scenes from CO3D (Reizenstein et al., 2021), RealEstate10K

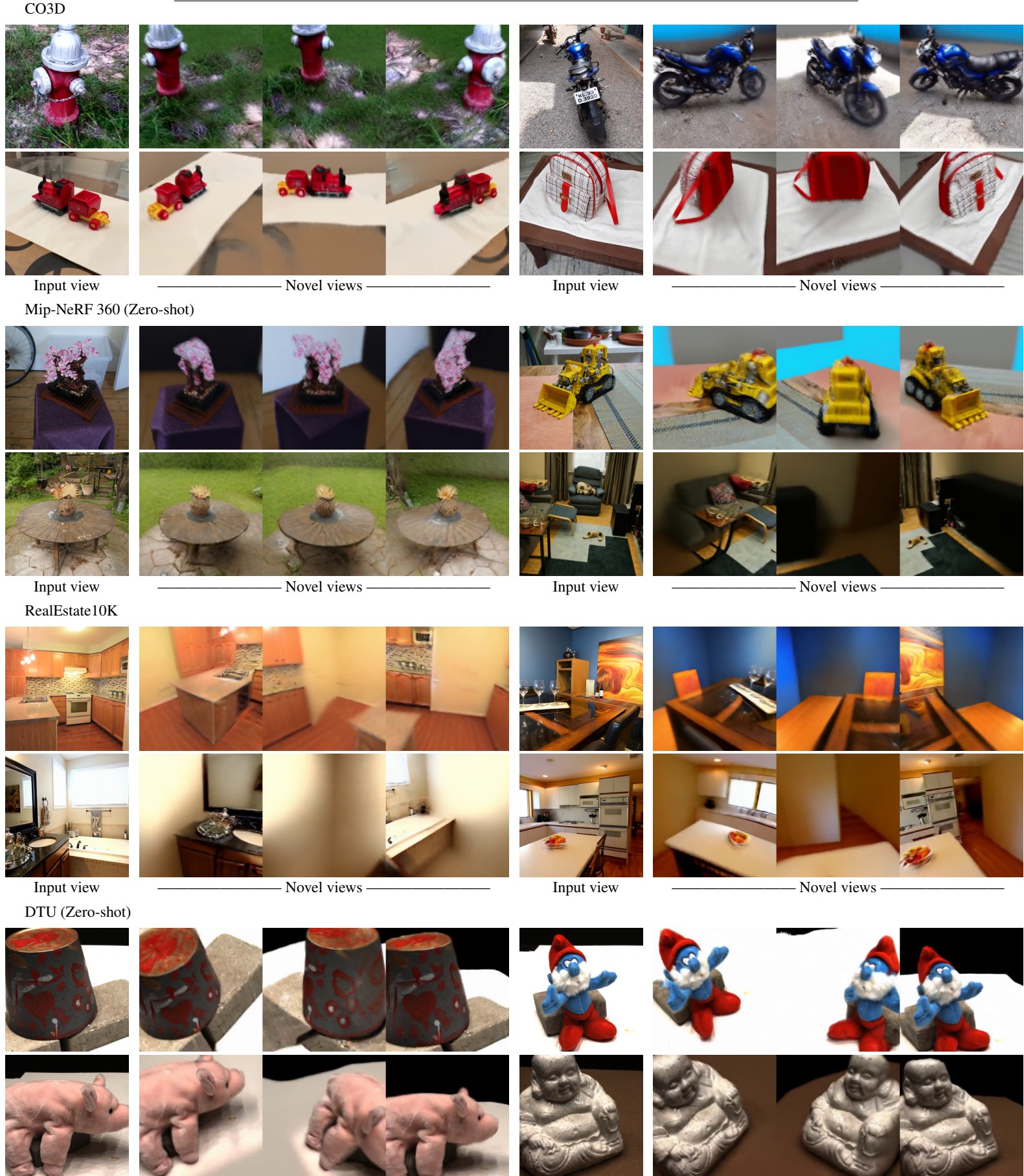

Figure 1: Results for view synthesis from a single image. All NeRFs are predicted by the *same* model.

(Zhou et al., 2018), and ACID (Liu et al., 2021), so that the model may potentially handle complex in-the-wild scenes. The mixed data of such scale and diversity are captured with a variety of camera settings and have several different types of 3D ground truth, e.g., computed with COLMAP (Schönberger & Frahm, 2016) or ORB-SLAM (Mur-Artal et al., 2015). We show that while the camera conditioning representations from prior methods (Liu et al., 2023b) are too ambiguous or inexpressive to model in-the-wild scenes, our new camera parametrization and normalization scheme allows exploiting such diverse data sources and leads to superior NVS on real-world scenes.

Building a 2D conditional diffusion model that works effectively for in-the-wild scenes enables us to then study the limitations of SDS in the scene setting. In particular, we observe limited diversity from SDS in the generated scene backgrounds when synthesizing long-range (e.g., 180-degree) novel views. We therefore propose "SDS anchoring" to ameliorate the issue. In SDS anchoring, we propose to first sample several "anchor" novel views using the standard Denoising Diffusion Implicit Model (DDIM) sampling (Song et al., 2021). This yields a collection of pseudo-ground-truth novel views with diverse contents, since DDIM is not prone to mode collapse like SDS. Then, rather than using these views as RGB supervision, we sample from them randomly as conditions for SDS, which enforces diversity while still ensuring 3D-consistent view synthesis.

ZeroNVS achieves strong zero-shot generalization to unseen data. We set a new state-of-the-art LPIPS score on the challenging DTU benchmark, even outperforming methods that were directly fine-tuned on this dataset. Since the popular benchmark DTU consists of scenes captured by a forward-facing camera rig and cannot evaluate more challenging pose changes, we propose to use the Mip-NeRF 360 dataset (Barron et al., 2022) as a single-image novel view synthesis benchmark. ZeroNVS achieves the best LPIPS performance on this benchmark. Finally, we show the potential of SDS anchoring for addressing diversity issues in background generation via a user study. To summarize, we make the following contributions:

- We propose ZeroNVS, which enables full-scene NVS from real images. ZeroNVS first demonstrates that SDS distillation can be used to lift scenes that are not object-centric and may have complex backgrounds to 3D.

- We show that the formulations on handling cameras and scene scale in prior work are either inexpressive or ambiguous for in-the-wild scenes. We propose a new camera conditioning parameterization and a scene normalization scheme. These enable us to train a single model on a large collection of diverse training data consisting of CO3D, RealEstate10K and ACID, allowing strong zero-shot generalization for NVS on in-the-wild images.

- We study the limitations of SDS distillation as applied to scenes. Similar to prior work, we identify a diversity issue, which manifests in this case as novel view predictions with monotone backgrounds. We propose SDS anchoring to ameliorate the issue.

- We show state-of-the-art LPIPS results on DTU *zero-shot*, surpassing prior methods fine-tuned on this dataset. Furthermore, we introduce the Mip-NeRF 360 dataset as a scene-level single-image novel view synthesis benchmark and analyze the performances of our and other methods. Finally, we show that our proposed SDS anchoring is overwhelmingly preferred for diverse generations via a user study.

## 2 RELATED WORK

**3D generation.** DreamFusion (Poole et al., 2022) proposed Score Distillation Sampling (SDS) as a way of leveraging a diffusion model to extract a NeRF given a user-provided text prompt. After DreamFusion, follow-up works such as Magic3D (Lin et al., 2023), ATT3D (Lorraine et al., 2023), ProlificDreamer (Wang et al., 2023), and Fantasia3D (Chen et al., 2023) improved the quality, diversity, resolution, or run-time. Other types of 3D generative models include GAN-based 3D generative models, which are primarily restricted to single object categories (Chan et al., 2021a; Niemeyer & Geiger, 2021; Gu et al., 2022; Chan et al., 2021b; Nguyen-Phuoc et al., 2019; Skorokhodov et al., 2022) or to synthetic data (Gao et al., 2022). Recently, 3DGP (Skorokhodov et al., 2023) adapted the

GAN-based approach to train 3D generative models on ImageNet. VQ3D (Sargent et al., 2023) and IVID (Xiang et al., 2023) leveraged vector quantization and diffusion on ImageNet, respectively.

**Single-image novel view synthesis.** PixelNeRF (Yu et al., 2021) and DietNeRF (Jain et al., 2021) learn to infer NeRFs from sparse views via training an image-based 3D feature extractor or semantic consistency losses, respectively. However, these approaches do not produce renderings resembling crisp natural images from a single image. Recent diffusion-based approaches achieve high quality results by separating the problem into two stages. First, a (potentially 3D-aware) diffusion model is trained, and second, the diffusion model is used to distill 3D-consistent scene representations given an input image via techniques like SDS (Poole et al., 2022), score Jacobian chaining (Wang et al., 2022), other diffusion-based guidance (Melas-Kyriazi et al., 2023; Deng et al., 2022a), or explicit 3D reconstruction from multiple sampled views of the diffusion model (Liu et al., 2023a;c). Unlike these works, ZeroNVS is trained on large real scene datasets and performs scene-level NVS. GenVS(Chan et al., 2023), a concurrent work, proposed a 3D-aware diffusion model based on 3D feature volumes. Though GenVS showed results on only a few types of in-domain data such as fire hydrants, ZeroNVS attains good performance on in-domain and out-of-domain scene categories.

## 3 APPROACH

We consider the problem of scene-level novel view synthesis from a single real image. Similar to prior work (Liu et al., 2023b; Qian et al., 2023), we first train a diffusion model $\mathbf{p}_\theta$ to perform novel view synthesis, and then leverage it to perform 3D SDS distillation. Unlike prior work, we focus on scenes rather than objects. Scenes present several unique challenges. First, prior works use representations for cameras and scale which are either ambiguous or insufficiently expressive for scenes. Second, the inference procedure of prior works is based on SDS, which has a known mode collapse issue and which manifests in scenes through greatly reduced background diversity in predicted views. We will address these challenges through improved representations and inference procedures for scenes compared with prior work (Liu et al., 2023b; Qian et al., 2023).

We shall begin by introducing some general notation. Let a scene $S$ consist of a set of images $X = \{X_i\}_{i=1}^n$, depth maps $D = \{D_i\}_{i=1}^n$, extrinsics $E = \{E_i\}_{i=1}^n$, and a shared field-of-view $f$. We note that an extrinsics matrix $E_i$ can be identified with its rotation and translation components, defined by $E_i = (E_i^R, E_i^T)$. We preprocess our data to consist of square images and assume intrinsics are shared within a scene, and that there is no skew, distortion, or off-center principal point. We will focus on the design of the conditional information which is passed to the view synthesis diffusion model $\mathbf{p}_\theta$ in addition to the input image. This conditional information can be represented via a function, $\mathbf{M}(D, f, E, i, j)$, which computes a conditioning embedding given the depth maps and extrinsics for the scene, the field of view, and the indices $i, j$ of the input and target view respectively. We learn a generative model over novel views $\mathbf{p}_\theta$ given by

$$X_j \sim \mathbf{p}_\theta(X_j | X_i, \mathbf{M}(D, f, E, i, j)) .$$

The output of $\mathbf{M}$ and the input image $X_i$ are the only information used by the model for NVS. As we illustrate in Figures 2, 3, 4 and 5, and verify in experiments, different choices for $\mathbf{M}$ can have drastic impacts on performance.

### 3.1 REPRESENTING OBJECTS FOR VIEW SYNTHESIS

Zero-1-to-3 (Liu et al., 2023b) represents poses with 3 degrees of freedom, given by an elevation $\theta$, azimuth $\phi$, and radius $z$. Let $\mathbf{P} : \mathrm{SE}(3) \to \mathbb{R}^3$ be the projection to this representation, then

$$\mathbf{M}_{\mathrm{Zero-1-to-3}}(D, f, E, i, j) = \mathbf{P}(E_i) - \mathbf{P}(E_j)$$

is the camera conditioning representation used by Zero-1-to-3. For object mesh datasets such as Objaverse (Deitke et al., 2022) and Objaverse-XL (Deitke et al., 2023), this representation is appropriate because the data is known to consist of single objects without backgrounds, aligned and centered at the origin and imaged from training cameras generated with three degrees of freedom. However, such a parameterization limits the model's ability to generalize to non-object-centric images, and to real-world data. In real-world data, poses can have six degrees of freedom, incorporating both rotation (pitch, roll, yaw) and 3D translation. An illustration of a failure of the 3DoF camera representation due to the camera's roll is shown in Figure 2.

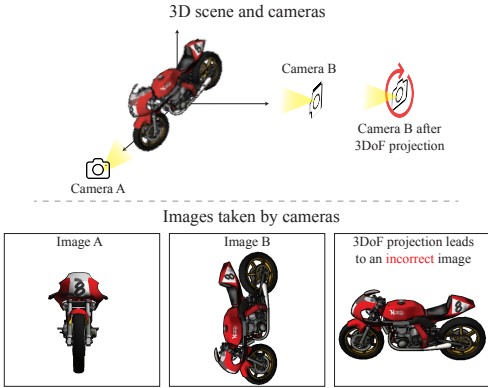

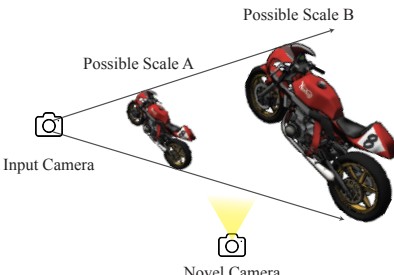

Figure 2: A 3DoF camera pose captures camera elevation, azimuth, and radius for a camera pointed at the origin but is incapable of representing a camera's roll (pictured) or cameras oriented arbitrarily in space.

Figure 3: To a monocular camera, a small object close to the camera (left) and a large object at a distance (right) appear identical, despite representing different scenes. Scale ambiguity in the input view causes ambiguity in NVS.

## 3.2 REPRESENTING GENERIC SCENES FOR VIEW SYNTHESIS

For scenes, we should use a camera representation with six degrees of freedom that can capture all possible positions and orientations. One straightforward choice is the relative pose parameterization (Watson et al., 2023). We propose to also include the field of view as an additional degree of freedom. We term this combined representation "6DoF+1". This gives us

$$\mathbf{M}_{6\text{DoF}+1}(D, f, E, i, j) = [E_i^{-1} E_j, f].$$

Importantly, $\mathbf{M}_{6\text{DoF}+1}$ is invariant to any rigid transformation $\tilde{E}$ of the scene, so that we have

$$\mathbf{M}_{6\text{DoF}+1}(D, f, \tilde{E} \cdot E, i, j) = [(\tilde{E} \cdot E_i)^{-1} \tilde{E} \cdot E_j, f] = [E_i^{-1} E_j, f] .$$

This is useful given the arbitrary nature of the poses for our datasets which are determined by COLMAP or ORB-SLAM. The poses discovered via these algorithms are not related to any semantically meaningful alignment of the scene, such as a rigid transformation and scale transformation which align the scene to some canonical frame and unit of scale.

Although we have seen that $\mathbf{M}_{6\text{DoF}+1}$ is invariant to rigid transformations of the scene, it is not invariant to scale. The scene scales determined by COLMAP and ORB-SLAM are also arbitrary, and in practice may vary by orders of magnitude. One solution is to simply normalize the camera locations to have, on average, the unit norm when the mean of the camera locations is chosen as the origin. Let $\mathbf{R}(E, \lambda) : \text{SE}(3) \times \mathbb{R} \rightarrow \text{SE}(3)$ be a function that scales the translation component of the extrinsic matrix $E$ by $\lambda$. Then we define

$$s = \frac{1}{n} \sum_{i=1}^{n} \| E_i^T - \frac{1}{n} \sum_{j=1}^{n} E_j^T \|_2 ,$$

$$\mathbf{M}_{6\text{DoF}+1,\ \text{norm.}}(D, f, E, i, j) = \left[ \mathbf{R}\left(E_i, \frac{1}{s}\right)^{-1} \mathbf{R}\left(E_j, \frac{1}{s}\right), f\right] ,$$

where $s$ is the average norm of the camera locations when the mean of the camera locations is chosen as the origin. In $\mathbf{M}_{6\text{DoF}+1,\ \text{norm.}}$, the camera locations are normalized via rescaling by $\frac{1}{s}$, in contrast to $\mathbf{M}_{6\text{DoF}+1}$ where the scales are arbitrary. This choice of $\mathbf{M}$ assures that scenes from our mixture of datasets will have similar scales.

## 3.3 ADDRESSING SCALE AMBIGUITY WITH A NEW NORMALIZATION SCHEME

The representation $\mathbf{M}_{6\text{DoF}+1,\ \text{norm.}}$ achieves reasonable performance on real scenes by addressing issues in prior representations with limited degrees of freedom and handling of scale. However,

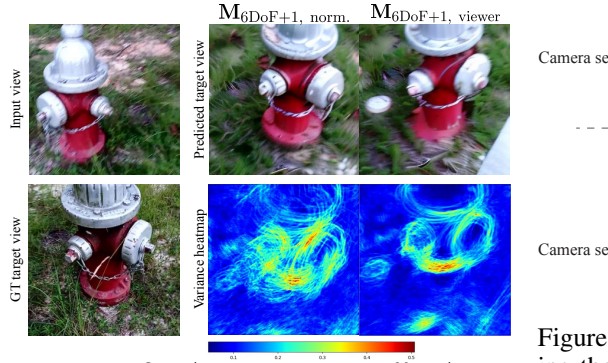

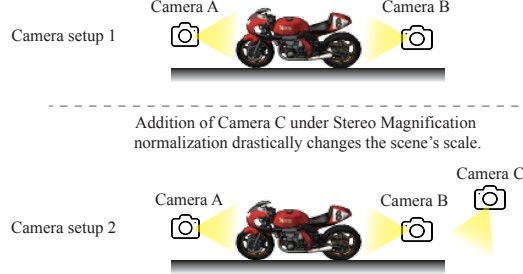

Figure 4: Samples and variance heatmaps of the Sobel edges of multiple samples from ZeroNVS. $\mathbf{M}_{6\text{DoF}+1, \text{ viewer}}$ reduces randomness from scale ambiguity.

Figure 5: Top: A scene with two cameras facing the object. Bottom: The same scene with a new camera added facing the ground. Aggregating the cameras before computing the scale leads to different scales for each setup. Our proposed viewer-centric normalization avoids this.

a normalization scheme that better addresses scale ambiguity may lead to improved performance. Scene scale is ambiguous given a monocular input image (Ranftl et al., 2022; Yin et al., 2022). This complicates NVS, as we illustrate in Figure 3. We therefore choose to condition on the scale by introducing information about the scale of the visible content to our conditioning embedding function $\mathbf{M}$. Rather than normalize by camera locations, Stereo Magnification (Zhou et al., 2018) takes the 5-th quantile of each depth map of the scene, and then takes the 10-th quantile of this aggregated set of numbers, and declares this as the scene scale. Let $\mathbf{Q}_k$ be a function which takes the $k$-th quantile of a set of numbers, then we define

$$q = \mathbf{Q}_{10}(\{\mathbf{Q}_5(D_i)\}_{i=1}^n) ,$$

$$\mathbf{M}_{6\text{DoF}+1, \text{ agg.}}(D, f, E, i, j) = \left[ \mathbf{R}\left(E_i, \frac{1}{q}\right)^{-1} \mathbf{R}\left(E_j, \frac{1}{q}\right), f \right] ,$$

where in $\mathbf{M}_{6\text{DoF}+1, \text{ agg.}}$, $q$ is the scale applied to the translation component of the scene's cameras before computing the relative pose. In this way $\mathbf{M}_{6\text{DoF}+1, \text{ agg.}}$ is different from $\mathbf{M}_{6\text{DoF}+1, \text{ norm.}}$ because the camera conditioning representation contains information about the scale of the visible content from the depth maps $D_i$. Although conditioning with $\mathbf{M}_{6\text{DoF}+1, \text{ agg.}}$ improves performance, there are two issues. The first arises from aggregating the quantiles over all the images. In Figure 5, adding an additional Camera C to the scene changes the value of $\mathbf{M}_{6\text{DoF}+1, \text{ agg.}}$ despite nothing else having changed about the scene. This makes the view synthesis task from either Camera A or Camera B more ambiguous. To ensure this is impossible, we can simply eliminate the aggregation step over the quantiles of all depth maps in the scene. The second issue arises from different depth statistics within the mixture of datasets we use for training. ORB-SLAM generally produces sparser depth maps than COLMAP, and therefore the value of $\mathbf{Q}_k$ may have different meanings for each. We therefore use an off-the-shelf depth estimator (Ranftl et al., 2021) to fill holes in the depth maps. We denote the depth $D_i$ infilled in this way as $\bar{D}_i$. We then apply $\mathbf{Q}_k$ to dense depth maps $\bar{D}_i$ instead. We emphasize that the depth estimator is *not* used during inference or distillation. Its purpose is only for the model to learn a consistent definition of scale during training. These two fixes lead to our proposed normalization, which is fully viewer-centric. We define it as

$$q_i = \mathbf{Q}_{20}(\bar{D}_i) ,$$

$$\mathbf{M}_{6\text{DoF}+1, \text{ viewer}}(D, f, E, i, j) = \left[ \mathbf{R}\left(E_i, \frac{1}{q_i}\right)^{-1} \mathbf{R}\left(E_j, \frac{1}{q_i}\right), f \right] ,$$

where in $\mathbf{M}_{6\text{DoF}+1, \text{ viewer}}$, the scale $q_i$ applied to the cameras is dependent only on the depth map in the input view $\bar{D}_i$, different from $\mathbf{M}_{6\text{DoF}+1, \text{ agg.}}$ where the scale $q$ computed by aggregating over all $D_i$. At inference the value of $q_i$ can be chosen heuristically without compromising performance. Correcting for the scale ambiguities in this way improves metrics, which we show in Section 4.

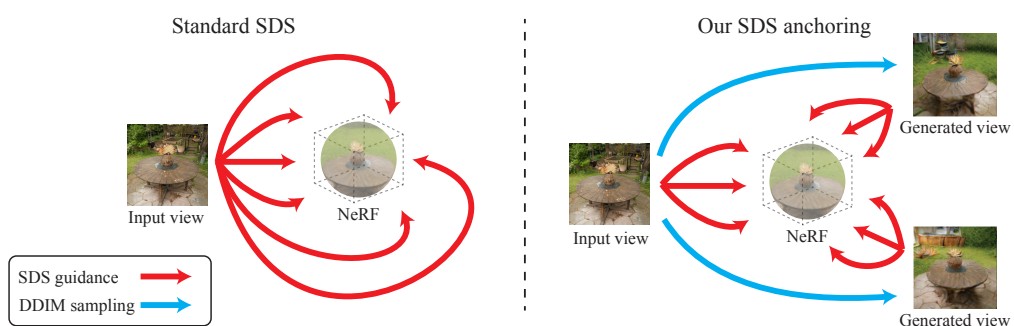

Figure 6: SDS-based NeRF distillation (left) uses the same guidance image for all 360 degrees of novel views. Our "SDS anchoring" (right) first samples novel views via DDIM (Song et al., 2020), and then uses the nearest image (whether the input or a sampled novel view) for guidance.

| NVS on DTU | LPIPS ↓ | PSNR ↑ | SSIM ↑ |
|---|---|---|---|
| DS-NeRF[†] | 0.649 | 12.17 | 0.410 |
| PixelNeRF | 0.535 | 15.55 | 0.537 |
| SinNeRF | 0.525 | **16.52** | **0.560** |
| DietNeRF | 0.487 | 14.24 | 0.481 |
| NeRDi | 0.421 | 14.47 | 0.465 |
| ZeroNVS (ours) | **0.414** | 12.76 | 0.460 |

Table 1: **SOTA comparison.** We set a new state-of-the-art for LPIPS on DTU despite being the only method not fine-tuned on DTU. † = Performance reported in Xu et al. (2022)

| NVS | LPIPS ↓ | PSNR ↑ | SSIM ↑ |
|---|---|---|---|
| **Mip-NeRF 360 Dataset** | | | |
| Zero-1-to-3 | 0.667 | 11.7 | 0.196 |
| PixelNeRF | 0.718 | **16.5** | **0.556** |
| ZeroNVS (ours) | **0.625** | 13.2 | 0.240 |
| **DTU Dataset** | | | |
| Zero-1-to-3 | 0.473 | 10.61 | 0.402 |
| PixelNeRF | 0.738 | 10.46 | 0.397 |
| ZeroNVS (ours) | **0.414** | 12.76 | 0.460 |

Table 2: **Zero-shot comparison**. Comparison with baselines trained on our mixture dataset.

### 3.4 IMPROVING DIVERSITY WITH SDS ANCHORING

Diffusion models trained with the improved camera conditioning representation $\mathbf{M}_{\text{6DoF+1, viewer}}$ achieve superior view synthesis results via 3D SDS distillation. However, for large viewpoint changes, novel view synthesis is also a generation problem, and it may be desirable to generate diverse and plausible contents rather than contents that are only optimal on average for metrics such as PSNR, SSIM, and LPIPS. However, Poole et al. (2022) noted that even when the underlying generative model produces diverse images, SDS distillation of that model tends to seek a single mode. For novel view synthesis of scenes via SDS, we observe a unique manifestation of this diversity issue: lack of diversity is especially apparent in inferred backgrounds. Often, SDS distillation predicts a gray or monotone background for regions not observed by the input camera.

To remedy this, we propose "SDS anchoring" (Figure 6). With SDS anchoring, we first directly sample $k$ novel views $\hat{\mathbf{X}}_k = \{\hat{X}_j\}_{j=1}^{k}$ with $\hat{X}_j \sim p(X_j|X_i, \mathbf{M}(D, f, E, i, j))$ from poses evenly spaced in azimuth for maximum scene coverage. We sample the novel views via DDIM (Song et al., 2020), which does not have the mode collapse issues of SDS. Each novel view is generated conditional on the input view. Then, when optimizing the SDS objective, we condition the diffusion model not on the input view, but on the nearest view . As shown quantitatively in Table 3 and qualitatively in Figure 9, SDS anchoring produces more diverse background contents. We provide more details about the setup of SDS anchoring in Appendix B.

## 4 EXPERIMENTS

### 4.1 SETUP

**Datasets.** Our models are trained on a mixture dataset consisting of CO3D, ACID, and RealEstate10K. Each example is sampled uniformly at random from the three datasets. We train at $256 \times 256$ resolution, center-cropping and adjusting the intrinsics for each image and scene as

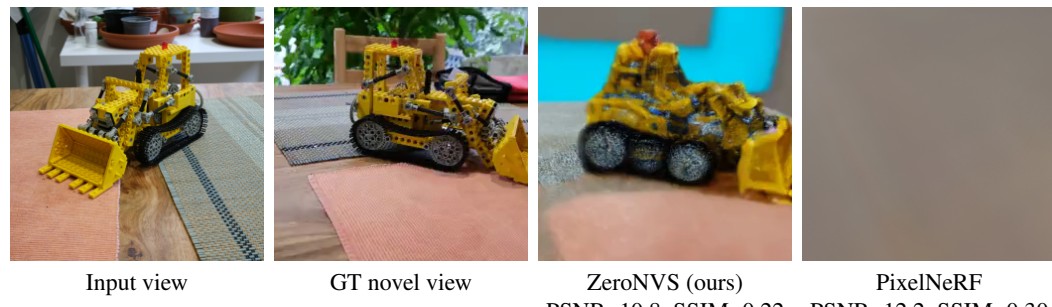

| Input view | GT novel view | ZeroNVS (ours) | PixelNeRF |
| | | PSNR=10.8, SSIM=0.22 | PSNR=12.2, SSIM=0.30 |

Figure 7: Limitations of PSNR and SSIM for view synthesis evaluation. Misalignments can lead to worse PSNR and SSIM values for predictions that are more semantically sensible.

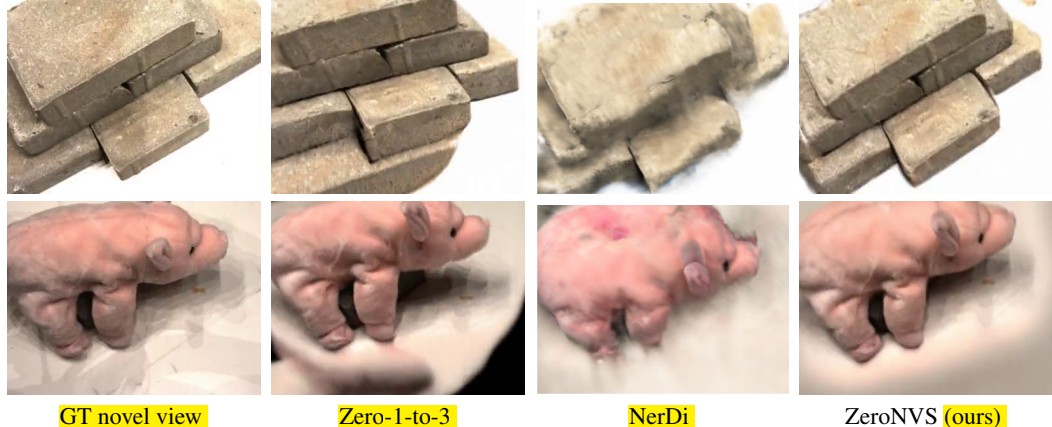

| GT novel view | Zero-1-to-3 | NerDi | ZeroNVS (ours) |

Figure 8: Qualitative comparison between baseline methods and our method.

necessary. We train using our representation $M_{6DoF+1, viewer}$ unless otherwise specified. We provide more training details in Appendix A. Our main evaluations are for zero-shot 3D consistent novel view synthesis, where we compare against other techniques on the DTU benchmark (Aanæs et al., 2016) and on the Mip-NeRF 360 dataset (Barron et al., 2022). We evaluate at $256 \times 256$ resolution except for DTU, for which we use $400 \times 300$ resolution to be comparable to prior art.

**Implementation details.** Our diffusion model training code is written in PyTorch and based on the public code for Zero-1-to-3 (Liu et al., 2023b). We finetune from Zero-1-to-3-XL. Our distillation code is implemented in Threestudio (Guo et al., 2023). For details, please consult Appendix B.

## 4.2 MAIN RESULTS

We evaluate all methods using the standard set of novel view synthesis metrics: PSNR, SSIM, and LPIPS. We weigh LPIPS more heavily in the comparison due to the well-known issues with PSNR and SSIM as discussed in (Deng et al., 2022a; Chan et al., 2023) and illustrated in Figure 7. The results are shown in Table 1. We first compare against baseline methods DS-NeRF (Deng et al., 2022b), PixelNeRF (Yu et al., 2021), SinNeRF (Xu et al., 2022), DietNeRF (Jain et al., 2021), and NeRDi (Deng et al., 2022a) on DTU. All these methods are trained on DTU, but we achieve a state-of-the-art LPIPS despite being fully zero-shot. We show visual comparisons in Figure 8. DTU scenes are limited to relatively simple forward-facing scenes. Therefore, we introduce a more challenging benchmark dataset, the Mip-NeRF 360 dataset, to benchmark the task of 360-degree view synthesis from a single image. We use this benchmark as a zero-shot benchmark, and train three baseline models on our mixture dataset to compare zero-shot performance. Our method is the best on LPIPS for this dataset. On DTU, we exceed Zero-1-to-3 and the zero-shot PixelNeRF model on all metrics, not just LPIPS. Performance is shown in Table 2.

Limited diversity is a known issue with SDS-based methods, but the long run time of SDS-based methods makes typical generation-based metrics such as FID cost-prohibitive. Therefore, we quan-

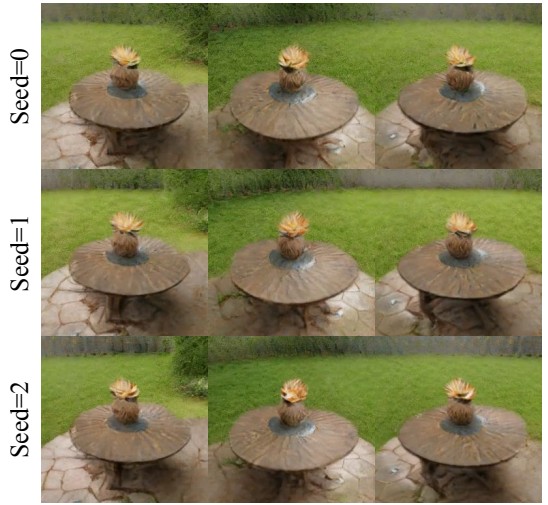 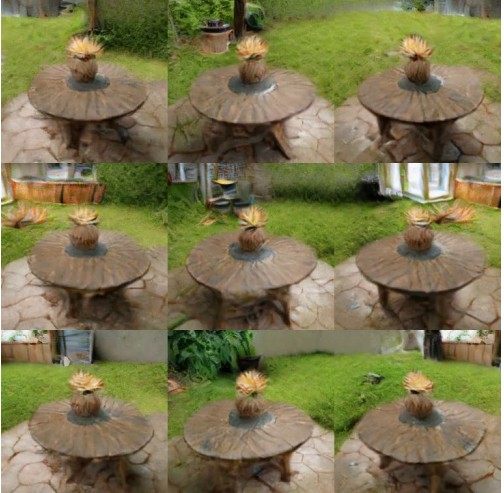

Novel views (standard SDS)      Novel views (SDS anchoring)

Figure 9: Qualitative depiction of the effects of SDS anchoring. Whereas standard SDS (left) tends to predict monotonous backgrounds, our SDS anchoring (right) generates more diverse backgrounds.

| User study | % that prefer SDS anchoring |
|---|---|
| Realism | 78% |
| Creativity | 82% |
| Overall | 80% |

Table 3: **The effectiveness of SDS anchoring.** Users prefer SDS anchoring.

| NVS on DTU | LPIPS ↓ | PSNR ↑ | SSIM ↑ |
|---|---|---|---|
| All datasets | **0.421** | **12.2** | **0.444** |
| -ACID | 0.446 | 11.5 | 0.405 |
| -CO3D | 0.456 | 10.7 | 0.407 |
| -RealEstate10K | 0.435 | 12.0 | 0.429 |

Table 4: **Ablation study on training data.** Training on all datasets improves performance.

tify the improved diversity from using SDS anchoring via a user study on the Mip-NeRF 360 dataset. The results, shown in Table 3, show a strong human preference for the more diverse scenes generated via SDS anchoring. In addition, Figure 9 includes qualitative examples of SDS anchoring.

We verify the benefits of each of our multiple multiview scene datasets in Table 4. In Table 5, we analyze the diffusion model's performance on held-out subsets of our datasets, with the various parameterizations discussed in Section 3. Due to computational constraints, we train the ablation diffusion models for fewer steps than our main model, hence the slightly worse performance relative to Table 1. We provide more details in Appendix C.

| | 2D novel view synthesis | | | | | | | | | 3D NeRF distillation | | |
|---|---|---|---|---|---|---|---|---|---|---|---|---|
| | CO3D | | | RealEstate10K | | | ACID | | | DTU | | |
| Conditioning | PSNR | SSIM | LPIPS | PSNR | SSIM | LPIPS | PSNR | SSIM | LPIPS | PSNR | SSIM | LPIPS |
| $\mathbf{M}_{\text{Zero-1-to-3}}$ | 12.0 | .366 | .590 | 11.7 | .338 | .534 | 15.5 | .371 | .431 | 10.3 | .384 | .477 |
| $\mathbf{M}_{\text{6DoF+1}}$ | 12.2 | .370 | .575 | 12.5 | .380 | .483 | 15.2 | .363 | .445 | 9.5 | .347 | .472 |
| $\mathbf{M}_{\text{6DoF+1, norm.}}$ | 12.9 | .392 | .542 | 12.9 | .408 | .450 | 16.5 | .398 | .398 | 11.5 | .422 | .421 |
| $\mathbf{M}_{\text{6DoF+1, agg.}}$ | 13.2 | .402 | .527 | **13.5** | **.441** | .417 | 16.9 | .411 | .378 | **12.2** | .436 | **.420** |
| $\mathbf{M}_{\text{6DoF+1, viewer}}$ | **13.4** | **.407** | **.515** | **13.5** | .440 | **.414** | **17.1** | **.415** | **.368** | **12.2** | **.444** | .421 |

Table 5: **Ablation study on the conditioning representation M.** Our conditioning representation ($\mathbf{M}_{\text{6DoF+1, viewer}}$) matches or outperforms other conditioning representations.

## 5 CONCLUSION

We have introduced ZeroNVS, a system for 3D-consistent novel view synthesis from a single image for generic scenes. We showed its state-of-the-art performance on existing NVS benchmarks and proposed the Mip-NeRF 360 dataset as a more challenging benchmark for single-image NVS.

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
