# OpenReview forum: "ZeroNVS: Zero-shot 360-degree View Synthesis from a Single Real Image"
_ICLR.cc/2024/Conference — ICLR 2024 Conference Withdrawn Submission_

### Official Review · Reviewer_1pBn · 2023-10-27

**Soundness:** 2 fair
**Presentation:** 3 good
**Contribution:** 2 fair
**Rating:** 5
**Confidence:** 4

**Summary:**

This paper tackles single-image novel view synthesis for in-the-wild scenes with diffusion models. Its main contributions lie in three folds.

1) The authors train a single-view-to-3D model for scene with complex backgrounds.
2) To make use of diverse datasets against scale ambiguities, the authors use off-the-shelf depth estimation to align the scales.
3) The authors adopt SDS anchoring which conditions the diffusion model on the nearest view in score distillation.

The model is trained on a mixture of CO3D, ACID, and RealEstate10K and evaluated on DTU and Mip-NeRF 360 datasets, achieving SOTA on both of them. The qualitative results and user study have also shown promising results.

**Strengths:**

1. The quantitative results are better than current state-of-the-art approaches.
2. The analysis of SDS anchoring is useful in scene-based single-image novel view synthesis.

**Weaknesses:**

1. The qualitative results are lack of comparison with other methods. This makes readers hard to decide if the qualitative results are better. Most of the results shown in the paper are still object-centric while with a complex background. It would be good to show comparisons with object-centric methods that mask out the background to see if the novel view synthesis for the object is better.
2. The authors mentioned SDS anchoring can increase diversity but did not include any diverse results to support this point, only showing a user study which is not convincing enough. It would also be good to use FID metrics in comparison.
3. The technical contribution is limited. The 2nd contribution (aligning the scale of different datasets) is weak to me. And the 3rd contribution (SDS anchoring) is not well supported.

**Questions:**

1. In Sec. 3.2 "be a function which scales the translation component of the extrinsic matrix $E$ by $\lambda$" and the following equation - is the L2 norm applied to the whole extrinsic matrix or only the translation part?
2. For SDS anchoring if we condition the diffusion model on a generated view, is it possible to degenerate?

---

> ### Author Response · Authors · 2023-11-15
> **Reply to 1pBn**
>
> Thanks for your comments. We will address the weaknesses and questions below.
> * **Qualitative results**. We agree that more qualitative comparisons are needed, and we include them in new Figures (Figures 8 and 9) and also in the revised supplementary website. Object-centric methods which can mask out the background achieve good results in object centric scenes if the background is not considered. However, the focus of ZeroNVS is to also handle complex scenes with background, such as the many RealEstate10K, Mip-NeRF-360 and DTU scenes in the supplementary material.
> * **SDS anchoring**. We added a new figure (Figure 9) in the main paper which illustrates results with and without SDS anchoring. We also include a gallery of video results with and without SDS anchoring in the supplementary website.
> * **On technical contribution**. We hope that the additional support for SDS anchoring which we have discussed above is helpful. Regarding the handling of the scale, Table 2 shows that Zero-1-to-3 cannot appropriately model scenes without our proposed camera conditioning representations when trained on the same data. Table 5 demonstrates the value of each of our successive improvements to the camera conditioning representation.
> * **L2 norm question**. Only to the translation part.
> * **Degeneration during SDS anchoring**. In practice we observe that this does not happen. It is possible that attempting to do this sort of procedure autoregressively would eventually lead to degeneration, but the novel view DDIM samples are sampled once from the input image at the start of training and are held fixed during training, so there is no possibility for them to degenerate. For examples of how SDS anchoring can improve background quality, please consult Figure 9 and also the revised supplementary website.

---

> > ### Author Response · Authors · 2023-11-17
> > **Additional questions?**
> >
> > We hope our response addresses your questions; if so, we would appreciate it if you would reconsider your score accordingly.  Please let us know soon if you have additional questions or concerns.

---

### Official Review · Reviewer_3tNs · 2023-10-29

**Soundness:** 2 fair
**Presentation:** 3 good
**Contribution:** 3 good
**Rating:** 3
**Confidence:** 4

**Summary:**

This paper approaches the task of single image novel view synthesis for scenes. It proposes to finetune Zero 123 on a collection of three large datasets (CO3D, RE10K, ACID) and generalize to evaluation datasets (Mip-NeRF 360, DTU) zero-shot. In order to train on data with 6DoF data, it adapts the existing pose parameterization of Watson et al. To train on data with scale ambiguity, it uses a normalization scheme; to encourage diversity it proposes SDS-anchoring. It outperforms reported baselines in LPIPS on DTU and MipNeRF-360.

**Strengths:**

Introduces new training paradigm which yields respectable results
- Technical contributions of handling changing depth scale and intrinsics allow the model to train on several large datasets.
- The model generalizes well to new data zero-shot
- Conditioning representation is ablated
SDS-anchoring is effective in improving sampled view diversity
- Clear from qualitative and quantitative results. Idea is intuitive and novel.
Paper is generally clear and well-motivated
- Fig 2-5 are effective along with the text, in motivating the major contributions of the paper (excluding the motivation for the choice of data)

**Weaknesses:**

Summary: I believe the paper has good contributions and presentation, but is currently missing several important evaluations. If the paper had more robust and convincing experiments I would be open to raising my score.

Missing comparisons
- The paper claims SSIM and PSNR are not appropriate for views the model is hallucinating. It is true these are not the best metrics, but it then only reports one metric (LPIPS), and contains only one visual comparison against prior work (Figure 7) – this includes supplemental and website. In a task of novel view synthesis, if two out of the three reported metrics are weaker than prior work, visual results are necessary to convince reviewers the proposed model improves on results. More examples would also be helpful to understand the drop in PSNR and SSIM numbers.
- The paper says “long runtime makes typical generation-based metrics such as FID cost-prohibitive”. I’m confused: in my experience, FID can be run in a minute or two over thousands of images. Given FID is an important metric for hallucinated images, it (or perhaps KID, or similar) should be reported at least on a subset of images. The alternative could be a human A/B test, which is not as ideal, since it cannot be precisely replicated, but still could give more defense to the results vs. existing methods.
- PixelNeRF is not the state of the art on single-image novel view synthesis. Table 2 is central to the paper’s argument, but unfortunately the comparisons are limited to Zero-1-to-3 and PixelNeRF.
    - The proposed method finetunes this on scenes, so should improve over Zero-1-to-3; so it would be good to see other baselines as well.
    - These two baselines do not have mechanisms to deal with differing intrinsics. Perhaps a better comparison would be to train these only on e.g. CO3D or only on RealEstate10K
    - As a result, it is hard to determine how effective the model is vs. prior work, or of the main contribution is training on large data?
    - Perhaps some recent SOTA methods could be trained on the same data. For instance (comparison does not have to be to this baseline, just an idea:) Consistent View Synthesis with Pose-Guided Diffusion Models (Tseng et al., CVPR 23), SynSin (Wiles et al., CVPR 20)
    - RegNeRF is better than DietNeRF in 3, 6 and 9 view. It should be considered: (RegNeRF: Regularizing Neural Radiance Fields (Niemeyer et al., CVPR 22))
-	Can the proposed model be finetuned e.g. on DTU for performance gain? If not, why not?

**Questions:**

- Why train on ACID? I was under the impression this doesn’t have views facing 360 degrees, but is rather mostly forward trajectory. What was the motivation for these datasets?
- Ablations were trained for fewer steps (25k) for compute reasons. How much compute was used? How long did training take?
- Typo, Figure 6 caption

---

> ### Author Response · Authors · 2023-11-15
> **Response to 3tNs**
>
> Thanks for your comments. We address the questions below.
> * **Metrics and qualitative comparisons**. We agree that more qualitative comparisons are needed, and we have included far more extensive qualitative comparisons in the revised version. They are in the new Figures (Figures 8 and 9) and also in the revised supplementary website.
> * **Runtime of FID**. We regret the miscommunication – we did not mean that the runtime of FID was too high. Rather, FID requires thousands of samples, and the runtime of SDS distillation-based methods such as DreamFusion and our method for generating one sample is on the order of hours. Thus, FID is impractical in this case, and we rely on view synthesis metrics on small datasets like DTU and Mip-NeRF 360 and user studies for our analysis.
> * **Fair comparison to baselines**. We first wish to clarify an apparent misunderstanding. For Table 2, we do not compare against the released weights for Zero-1-to-3 and PixelNeRF, but rather compare against retrained versions which are trained with identical data to ZeroNVS for a fair comparison, as we indicated in the table caption and text in the experiment section. Thus, for Table 2 the only difference between Zero-1-to-3 and ZeroNVS is the novelties we proposed for conditioning in our approach section, as the training data is identical. These novelties are what lead to the significant performance gains.
> * **Comparison to SOTA**. Please note that in our Table 1 and Table 2, we compare to a diverse set of recent methods, including DS-NeRF, PixelNeRF, SinNeRF, DietNeRF, NeRDi, Zero-1-to-3, and PixelNeRF. Since Zero-1-to-3 and NeRDi are SOTA approaches for view synthesis, we consider the combination of Table 1 and Table 2 a fair evaluation against existing SOTA.
> * **Can the model be finetuned on DTU?** The model can be finetuned on DTU and this would likely further improve performance. We chose not to do this as our performance was already SOTA for LPIPS and we preferred to emphasize the zero-shot nature of the model. We will finetune ZeroNVS and DTU and report the performance in a revised version. Due to the limited time, this may not be ready before the end of the rebuttal period.
> * **ACID**. Although ACID contains many examples of forward trajectories, this is still a useful signal for SDS distillation. Since ZeroNVS is trained on lots of examples of small perturbations of a forward-facing camera, this helps the scene geometry to quickly converge early in SDS distillation even when the sampled cameras are still mostly forward-facing and near the input camera, due to the diffusion model’s prior for parallax. Further support for this argument can be found in Table 4, where training the model without ACID data results in worse performance on DTU.
> * **Ablations**. Our diffusion models require an 8xA100 machine to train, which is the same as Zero-1-to-3. Training one diffusion model for 60,000 steps takes approximately a week, and since our ablations necessitated training 8 unique models (four in Table 4, and four additional unique models in Table 5), we trained these models for fewer steps. We provide more training details in Appendix A of the supplementary material.
> * **Typo**. We have fixed this and various typos in the revised version.

---

> > ### Author Response · Authors · 2023-11-17
> > **Additional questions?**
> >
> > We hope our response addresses your questions; if so, we would appreciate it if you would reconsider your score accordingly.  Please let us know soon if you have additional questions or concerns.

---

### Official Review · Reviewer_3848 · 2023-10-30

**Soundness:** 3 good
**Presentation:** 2 fair
**Contribution:** 3 good
**Rating:** 6
**Confidence:** 3

**Summary:**

**Summary:**
This paper introduces ZeroNVS, representing a scene-level NVS approach. In addition, it proposes several conditioning representation methods.

**Strengths:**

**Advantages:**
1. The paper gives a comprehensive introduction to the conditioning representation method.
2. Under the zero-shot setting, the result is good,  but there is still a significant potential for further improvement.
3. The overall writing is satisfactory but could benefit from further refinement.

**Weaknesses:**

**Disadvantages:**
1. The contributions made in the paper appear nuanced, making it challenging to discern the most crucial contribution.
2. It would be beneficial to compare the proposed method with GenNVS, which demonstrates leading performance in scene-central applications.

**Questions:**

**Questions/Concerns:**
1. What distinguishes ZeroNVS from Zero123 when claiming that your method is scene-central as opposed to object-central? It is unclear which elements bolster this claim. Specifically, does the distinction lie in Zero123 being trained on Objectverse while your method relies on RealEstate10K, ACID, and CO3D?
2. Regarding the zero-shot application on the DTU Dataset, how does ZeroNVS stack up against GenNVS?
Moreover, if ZeroNVS undergoes fine-tuning on the DTU dataset, how does its performance compare to that of GenNVS?
Notably, in a zero-shot setting, it lags behind GenNVS, which has been trained on the DTU dataset.
3. About the comparison with Zero123, can you further report the result of ZeroNVS on GSO dataset?

**Reference:**
@misc{chan2023genvs,
  title={GeNVS: Generative novel view synthesis with 3D-aware diffusion models},
  author={Chan, Eric R and Nagano, Koki and Chan, Matthew A and Bergman, Alexander W and Park, Jeong Joon and Levy, Axel and Aittala, Miika and De Mello, Shalini and Karras, Tero and Wetzstein, Gordon},
  year={2023},
  publisher={arXiv}
}

**Details Of Ethics Concerns:**

see above

---

> ### Author Response · Authors · 2023-11-15
> **Reply to 3848**
>
> Thanks for your comments. We address the questions below.
> - **Clarification on Contributions**. We have substantially revised our introduction and approach to more clearly delineate the contributions, namely
>     - A model, ZeroNVS, which solves the challenging problem of zero-shot single-image 360-degree view synthesis, described in the Approach section and with extensive results provided;
>     - New camera conditioning representations, which enable training on large-scale mixed datasets of real scenes, as supported in Table 5;
>     - SDS anchoring to address the limitations of SDS as applied to scenes, as supported in Table 3 and Figure 9;
>     - Extensive quantitative analysis and SOTA performance, as supported in Tables 1 and 2.
> - **Comparison with GenVS**. We have now discussed GenVS as concurrent work in the related work section. Please note that GenVS has not released the code yet. Please also note that GenVS was presented at ICCV (Oct. 2-6), whereas the ICLR deadline was at the end of September.
> - **Differences from Zero123**: The key differences between our method and Zero-1-to-3 are the camera conditioning representations we develop in the approach section, and our proposed system for SDS anchoring. We show in Table 2 that Zero-1-to-3, *even when trained on the same exact dataset as ZeroNVS*, achieves far worse performance. This is due to the more expressive camera conditioning representations which enable training on real-world scene data, a claim we support in Table 5.
> - **Result of ZeroNVS on GSO dataset**. GSO does not consist of real-world scenes, but rather is a synthetic dataset of object-centric scenes on white backgrounds, thus out of distribution of ZeroNVS which targets real images with potentially complex backgrounds. Instead, we compare against Zero-1-to-3 by training both Zero-1-to-3 and ZeroNVS on the same data, and measuring the performance on DTU and Mip-NeRF 360 datasets in Table 2. ZeroNVS significantly outperforms Zero-1-to-3.

---

> > ### Author Response · Authors · 2023-11-17
> > **Additional questions?**
> >
> > We hope our response addresses your questions; if so, we would appreciate it if you would reconsider your score accordingly.  Please let us know soon if you have additional questions or concerns.

---

### Official Review · Reviewer_UiFn · 2023-10-31

**Soundness:** 3 good
**Presentation:** 2 fair
**Contribution:** 2 fair
**Rating:** 5
**Confidence:** 3

**Summary:**

This paper introduces a method for new view synthesis (NeRF reconstruction) from a single image. They propose a two-stage method, ZeroNVS, and new techniques to elevate single-image object-centric novel view synthesis (NVS) to NVS on real-world scenes. Comparing to its backbone method DreamFusion, the main contributions are:

- Introduce focal length into poses to address the scale ambiguity
- Improve SDS in DreamFusion to SDS-anchoring
- Adopt Multi-plane model in Stereo Magnification to do a scene scale normalization for better generalizations

**Strengths:**

- Introduce focal length into poses to address the scale ambiguity
- Improve SDS in DreamFusion to SDS-anchoring
- Adopt a Multi-plane model in Stereo Magnification to do a scene scale normalization for better generalizations

**Weaknesses:**

- Technical contributions are incremental, mostly are add-ons to previous methods
- What is DDIM sampling in Figure 6?  I feel it is the main content in SDS-anchoring, and possibly the key point of why SDS can be used to scenes not just objects. But no details and descriptions are found.
- Typos "us us to train..." in Section 3 (2nd paragraph).
- If PSNR and SSIM are not reliable, why not change to other metrics? One example in Figure 7 cannot show the whole picture, so if we should not trust them, why evaluating them in tables? Not to mention that in Table 2, ZeroNVS has the bect PSNR and SSIM in DTU dataset, is it good or not? It is very confusing here.

**Questions:**

See weakness

---

> ### Author Response · Authors · 2023-11-15
> **Reply to UiFn**
>
> Thanks for your comments. We would like to clarify that multi-plane models are not a part of our architecture. ZeroNVS predicts NeRFs from a single image. The confusion may have arisen from our use of the same camera normalization scheme as that in the baseline Stereo Magnification (Zhou et al, 2018).
>
> We address the questions below:
> * **Technical contributions**. The problem (scene-level 360-degree novel view synthesis from a single image) is a significant problem in computer vision, and demonstrating that it is possible with novel adaptations to existing techniques is a valuable contribution.
>
> * **On DDIM sampling**. We have added a citation and description in the draft. DDIM (Ho et al, 2020) is the sampling procedure for the 2D diffusion model. Different from SDS, which is a technique that allows one to use the diffusion model as a critic of novel views rendered by the NeRF, DDIM sampling generates 2D samples from the distribution of the diffusion model.
>
> * **Typo**. We have attached a revised manuscript in which various typos have been fixed.
>
> * **PSNR and SSIM**. Although we argue in Figure 7 and in the text that PSNR and SSIM should be weighted less compared with LPIPS, we still include PSNR and SSIM as this is standard in the view synthesis literature. As part of the rebuttal, we also include significantly more qualitative comparisons with baselines in our revised supplementary material, and added qualitative comparisons in Figures 8 and 9 of the main text. We hope that they can be consulted for a more comprehensive picture of the performance.

---

> > ### Author Response · Authors · 2023-11-17
> > **Additional questions?**
> >
> > We hope our response addresses your questions; if so, we would appreciate it if you would reconsider your score accordingly.  Please let us know soon if you have additional questions or concerns.

---

### Author Response · Authors · 2023-11-15
**General comment**

We thank all the reviewers for their thoughtful comments.

We have uploaded a revised version of the paper which has been substantially revised for clarity and for several of the suggested changes. Rewritten regions are highlighted in yellow. We have rewritten the introduction to more clearly identify the contributions as suggested by **3848**, streamlined the related work, and improved the approach and experiments sections. We added two new figures for qualitative comparisons (Figures 8 and 9), as suggested by **3tNs**, **1pBn**.  In addition, we have uploaded a revised supplementary material which contains even more qualitative video comparisons. We hope that these substantial efforts to clarify and improve the paper will more clearly explain our contributions and prevent any misunderstandings.

We respond to each reviewer’s specific concerns in our individual responses.